# The Recent Development of Multifunctional Gold Nanoclusters in Tumor Theranostic and Combination Therapy

**DOI:** 10.3390/pharmaceutics14112451

**Published:** 2022-11-14

**Authors:** Sisi Liu, Junyao Wang, Yuxin Song, Shuya He, Huaxin Tan

**Affiliations:** 1School of Public Health, Hengyang Medical School, University of South China, Hengyang 421001, China; 2Department of Biochemistry and Molecular Biology, School of Basic Medicine, Hengyang Medical School, University of South China, Hengyang 421001, China

**Keywords:** gold nanoclusters, cancer diagnosis, combination therapy

## Abstract

The rising incidence and severity of malignant tumors threaten human life and health, and the current lagged diagnosis and single treatment in clinical practice are inadequate for tumor management. Gold nanoclusters (AuNCs) are nanomaterials with small dimensions (≤3 nm) and few atoms exhibiting unique optoelectronic and physicochemical characteristics, such as fluorescence, photothermal effects, radiosensitization, and biocompatibility. Here, the three primary functions that AuNCs play in practical applications, imaging agents, drug transporters, and therapeutic nanosystems, are characterized. Additionally, the promise and remaining limitations of AuNCs for tumor theranostic and combination therapy are discussed. Finally, it is anticipated that the information presented herein will serve as a supply for researchers in this area, leading to new discoveries and ultimately a more widespread use of AuNCs in pharmaceuticals.

## 1. Introduction

The high incidence and mortality rate of cancer pose grave risks to the lives and well-being of all humans. It has long been a focus of research in life science to improve the accuracy of the early detection of malignant tumors and to address the dearth of effective tumor treatments [1]. With the rapid development of nanotechnology, the diversity of structures and functions of biological nanomaterials has been further enriched and spread at an alarming rate to life sciences and clinical medicine, especially new nanomaterials that integrate multiple modes of diagnostic and therapeutic strategies in one, making precise diagnosis and treatment integration and synergistic treatment possible, and this is eagerly anticipated around the globe [2,3].

Gold nanoparticles (AuNPs) are a type of colloidal or agglomerated particle with diameters between a few and hundreds of nanometers, composed of gold cores and surface shell layers. Due to their unique optical properties (surface plasmon resonance, surface-enhanced Raman scattering, etc.) and excellent catalytic properties, they hold great promise in a variety of applications, including biosensing, bioimaging, disease diagnosis, and treatment [4,5,6,7].

Gold nanoclusters (AuNCs) are gold nanomaterials with significantly smaller dimensions (≤3 nm) and typically comprise a few to tens of atoms [8]. Due to the quantum-limited effect, AuNCs have superior fluorescence properties and are utilized in a variety of scientific fields, including environmental detection, molecular labeling, and bioimaging [9,10,11,12,13]. In addition, because AuNCs are smaller than the renal threshold, they are easier to eliminate from the body than AuNPs, resulting in greater biosafety and in vivo application potential [14]. Physical, chemical, and biological techniques are now used by production enterprises and lab researchers to create AuNCs. In situ synthesis employing biomolecules (DNA, proteins, peptides, etc.) as templates are one of the chemical techniques that is gaining popularity among researchers [15,16,17]. The principal causes are as follows. Firstly, the biomolecular template contains numerous active functional groups, such as -SH, -COOH, -NH_2_, and -OH, which can bind gold atoms and improve their stability [18,19]. Secondly, some reducing amino acids (e.g., tryptophan, tyrosine) can reduce Au^3+^ ions to Au atoms in the presence of an appropriate pH environment, avoiding the use of strong reducing agents (e.g., NaBH4, CTAB) and have an improved biocompatibility [20]. Thirdly, the physical and chemical properties of AuNCs, such as the number of atoms, particle size, and optical properties, can be rapidly modified by adjusting the template amino acid or nucleotide sequences [21,22,23]. Lastly, the biological activities and functional binding sites of biomolecules provide a rich platform for further multi-functionalization of AuNCs [24,25].

Meanwhile, for tumor tissue enrichment, small AuNCs with high permeability and long retention are preferable. Surface-modified AuNCs can reduce the reticuloendothelial system (RES) and non-specific uptake, as well as specifically bind to overexpressed tumor cell receptors to enhance tumor cell accumulation, resulting in an enhanced cytotoxic effect against tumor cells [26,27]. The AuNCs can be rapidly excreted via the kidney, thereby minimizing damage to healthy tissues [14]. In comparison to large AuNPs, AuNCs possess a larger specific surface area and, consequently, greater surface energy. Due to this surface effect, the surface atoms of AuNCs are reactive and readily bondable with other atoms. Large payloads of drugs, genes, and other therapeutic molecules can be effectively trapped and protected from enzymatic degradation in complex physiological microenvironments [28,29,30]. Various internal and external stimuli may be used to regulate the release of drug-carrying molecules from functionalized AuNCs (e.g., pH, glutathione, light) [31,32]. As a result, they can be used as carriers for efficient targeted transport of therapeutic molecules, to enhance drug aqueous solubility, to prevent drug leakage in healthy tissues prematurely, and mitigate potential side effects.

Indeed, numerous reviews have been conducted on the design and application of AuNCs, particularly in terms of fluorescence imaging. Nonetheless, an increasing number of studies are currently attempting to fully integrate the various properties of AuNCs. So, from the perspective of integration and combined use of AuNCs for diagnosis and therapy, this review focuses on the combined therapies and cancer theranostics of AuNCs (Figure 1). Hence, this review focuses on the use of AuNCs as imaging, transport, and therapeutic agents for combined application. Specifically, in Section 2, we summarize the applications of AuNCs in fluorescence imaging, radiography, and other multimodal imaging for tumor theranostic. In Section 3, strategies for controlled release via tumor microenvironment response, photoactivation and nuclear targeting based on using AuNCs in constructing smart multifunctional vectors are discussed. Finally in Section 4, the combined therapies for cancer utilizing the radiosensitization and photothermal conversion effects of AuNCs are highlighted.

## 2. AuNCs as Imaging Agents in Tumor Theranostic

Since the successful construction of ultra-small AuNCs, the unique photoelectric effect resulting from their quantum size effect has been valued by researchers and utilized in a variety of sensing, detection, and bioimaging fields [33,34,35,36]. AuNCs are ideally suited for integrated medical applications in diagnostics and treatment due to their superior biocompatibility and functional versatility [37,38]. The atomic-level investigation of AuNCs has accelerated recently. Due to their precise size and composition, researchers have discovered that AuNCs have outstanding self-assembly and crystallization properties which endow them with more unique and diverse fluorescence properties [39,40]. To start, in this section, we summarize the recent studies on the integration and visualization of AuNCs for diagnosis and treatment based on different imaging modalities of AuNCs, respectively (Table 1).

### 2.1. Fluorescence Imaging

Biofluorescence imaging is a promising non-invasive, real-time, and high-resolution technique for the early detection of cancer [25,41]. Because of their superior biocompatibility and unique optical properties, AuNCs are ideal as fluorescent markers for bioimaging. Unlike metal nanoparticles with larger particle diameters, gold clusters have continuous or semi-continuous electronic energy levels, allowing them to have a more discrete electronic structure [42,43]. Their unique electronic structure gives them molecular-like properties, such as enhanced photoluminescence, electron transfer between the highest occupied molecular orbital (HOMO) and the lowest unoccupied molecular orbital (LUMO), efficient catalytic properties, magnetic properties, strong redox properties, and biocompatibility [44]. However, metal nanoclusters’ inherent low fluorescence quantum yield and weak cell penetration ability severely limit their use in practical applications [45,46]. Given that, common ways to improve the properties of AuNCs include ligand exchange, metal doping, aggregation-induced emission (AIE), and particle size modulation. These methods can be used to change the emission wavelength of AuNCs from ultraviolet to visible to near-infrared and to boost their fluorescence quantum yields [37,47,48,49]. Due to the near-infrared emission advantages, such as stronger tissue penetration, lower tissue absorption, high signal-to-noise ratio, and elimination of tissue self-fluorescence, these nanoclusters can meet the demand for real-time, in situ detection of organisms [50,51,52]. In 2010, Wang’s team achieved success in fluorescence imaging (FL) of tumors in mice using ultrasmall (~2.7 nm) near-infrared (NIR) AuNCs they had constructed [53]. Actually, the application of AuNCs for biofluorescence imaging has been the topic of a substantial number of articles and reviews, which we will not repeat in this paper.

On the other hand, fluorescence imaging can be used to identify tumor locations after AuNCs have been targeted and delivered to tumor cells. In this view, AuNCs can serve as carriers integrating diagnoses and treatments when they are introduced into the body. Taking advantage of MMP2 polypeptide (CPLGVRGRGDS) modification, Xia’s team constructed gold nanocluster nanoprobes loaded with photosensitizer chlorin e6 (Ce6) for near-infrared fluorescence imaging and lung cancer targeting therapy. The as-synthesized nanoprobes mark cancer cells more effectively than free Ce6 and enhance treatment efficacy. In addition, the presence of nanoprobe polyethylene glycol (PEG) shells improves biocompatibility and increases blood circulation when fluorescence imaging-guided malignancies are treated with photodynamic therapy (PDT) [54]. Li et al. recently reported a novel gold nanocluster-based combination therapeutic platform for pancreatic ductal adenocarcinoma (PDAC) that combines photothermal therapy (PTT)-carrier AuNCs with the ligand U11 peptide, which improves therapeutic platform internalization on pancreatic cancer cells. In vivo studies, the abdomen fluorescence signal was observed simultaneously with tail vein injection. The internal tumors receiving PDT/PTT combination therapy guided by fluorescence imaging were potently eradicated with negligible side effects on adjacent healthy tissues [55].

Additionally, the possibility to alter the AuNCs’ fluorescence properties permit the combination of disease therapy and multi-wavelength intelligent condition-responsive fluorescence imaging. For simultaneous in situ multilayer imaging with differing intracellular spatial distributions and fluorescence-guided photothermal treatment, Wang et al. have created a multifunctional nanomaterial-based all-in-one nanoplatform combining polydopamine (PDA)-loaded gold nanobipyramids and two different kinds of AuNC probes (Figure 1). In this elaborate design, AuNCs with different emission wavelengths were synthesized and tagged with two molecular probes, transmembrane glycoprotein mucin1 (MUC1) aptamer, and microRNA-21, respectively. The MUC1 aptamer labeled AuNCs detached from the nanocomposites upon the competitive combinations between MUC1 and its aptamer, turning on red fluorescence. Inside the cells. miRNA-21 hybridization with single-stranded DNA caused green fluorescence in AuNCs. In situ multilayer imaging of dual tumor biomarkers with distinct intracellular spatial distributions was obtained. Moreover, the photothermal effects of Au NBPs@PDA resulted in more effective cancer cell death, highlighting the all-in-one nanoplatforms potential for accurate detection and tumor treatment [56].

### 2.2. Radiography

Despite its widespread use, iodine compound has some drawbacks that make it less than ideal as a clinical contrast agent, including a short half-life, poor tumor targeting capabilities, and other issues [57,58,59]. Gold, on the other hand, has a high atomic number (Z = 79) and high electron density, resulting in a high X-ray attenuation coefficient [60]. Therefore, compared to the same concentration of iodine, AuNCs have a higher X-ray absorption coefficient, which is more beneficial for increasing the contrast between healthy and malignant regions in soft tissues. A growing number of AuNPs have been utilized successfully for X-ray and computed tomography (CT) imaging to date [61].

Although the advantages of smaller AuNCs in radiography are not as advantageous as those of larger AuNPs, the local plasmon resonance and larger cavity structure caused by the ultra-small size give AuNCs a greater photothermal conversion effect, allowing them to be used for both photothermal imaging and photothermal therapy [39,62,63,64,65,66]. This advantage of AuNCs makes it possible to combine radiography with photothermal imaging and photothermal therapy, a field of research that is currently in high demand. Wang et al. developed an active targeting nanosystem for gonadotropin-releasing hormone (GnRH) receptors on the surface of prostate cancer cells by modifying luteinizing hormone-releasing hormone analogue (LHRHa) antibodies on AuNCs. While enhancing the CT imaging effect, the adjustable optical and photothermal properties of AuNCs can be used to diagnose GnRH receptor-positive prostate cancer with dual-mode (FL/CT) imaging and PTT-targeted therapy in an integrated therapeutic approach [67]. Recently, Li et al. reported a HER2-modified thermosensitive liposome (immunoliposome)-assisted complex, named GTSL-CYC-HER2, by reducing AuNCs on its surface. The HER2 modification not only facilitated targeting the breast cancer tumors but also catalyzed the gold nanocluster shells in situ. Under NIR irradiation, thermosensitive liposomes released the loaded anti-tumor agent cyclopamine to destroy the tumor-associated matrix. As a result, the size-tunable gold-wrapped immunoliposome was more likely to infiltrate deeper levels of the tumor, while the presence of AuNCs enables GTSL-CYC-HER2 multimodal imaging and synergistic therapy. In this design, AuNCs not only serve as contrast agents for CT and photothermal (PT) imaging, but also enable synergistic photothermal and chemotherapeutic treatment because of their superior photothermal conversion effect, which results from the local surface plasma resonance and larger cavity structure of ultra-small AuNCs [68].

### 2.3. Multi-Modal Imaging

In addition to the common fluorescence imaging, photothermal imaging, and as a contrast agent to participate in disease diagnosis and treatment, the controllable size of AuNCs, and easily modified surface properties, can be combined with a variety of materials or biomolecules with different imaging rationale and therapeutic functions through intelligent design to achieve the multimodal imaging and therapeutic combination of AuNCs [37,69]. Mei et al. loaded AuNCs and chlorine e6 (photosensitizer, Ce6) onto the surface of Gd-doped layered double hydroxide (Gd-LDH) monolayer nanosheets. The results of in vitro studies showed that the loading of AuNCs could lead to a much higher T1-weighted relaxivity (r1, the magnetic relaxation enhancement of the neighboring water protons, which determines MRI efficiency) greatly improved, indicating that the loading AuNCs could significantly improve the magnetic resonance imaging (MRI) sensitivity of Gd-LDH monolayer. Moreover, through the synergistic interaction between AuNCs and Gd-LDH, this treatment system has superior T1 MRI performance than the commercial MRI contrast agent Gd-DOTA [70]. Ma et al. loaded ionic liquid (IL), a microwave sensitizer, onto fluorescent AuNCs (BSA-AuNCs) with surface coupled Fe-metal organic framework nanoparticles (MIL-101(Fe) NPs) (Figure 2). The nanoparticles can be combined with magnetic resonance imaging (MRI) and fluorescence imaging for dual-mode imaging to accurately diagnose the specific location of the tumor. At the same time, microwave radiation can be used to kill tumor cells in the body by accelerating the production of ·OH from H_2_O_2_, which can be combined with microwave thermal therapy (MTT) and microwave enhancing dynamic therapy (MEDT) to improve the anti-tumor treatment effect [71].

In addition to bimodal imaging, gold nanocluster-based nanoenzymes can also be synthesized for combined therapy with trimodal imaging. Dan and his colleagues obtained ultramicro AuNCs-ICG nanoenzymes by further loading indocyanine green (ICG) through hydrophobic interaction and hydrogen bonding based on a biomineralization method. The results of in vivo experiments showed that the AuNCs carriers could protect the stable presence of ICG and prolong its circulation time, thus ensuring the full utilization of the inherent FL and photoacoustic (PA) imaging capabilities of ICG, while the high X-ray absorption properties of AuNCs enabled the ultrasmall AuNCs-ICG nanoenzyme to enhance CT imaging. The multimodal imaging (near-infrared fluorescence, photoacoustic, and computed tomography) abilities of this nanoenzyme can monitor and guide the combined treatment of PDT and radiotherapy (RT). Meanwhile, AuNCs-ICG nanoenzyme can effectively decompose intra-tumor H_2_O_2_ into O_2_ to regulate the tumor hypoxic environment and enhance the therapeutic effect of PDT and RT on cancer [72].

## 3. AuNCs as Transport Agents in Combined Therapy

AuNCs are widely used for drug delivery and controlled release in vivo and ex vivo as one of the metallic nanomaterials with the longest research history. As one of the special ultra-small size nanostructures, AuNCs have a greater potential for combinatorial applications [73]. Initially, AuNCs have a stable and inert internal core that can shield encapsulated drug molecules. Further, AuNCs have a high surface to volume ratio and can be loaded with a substantial quantity of small-molecule drugs via reasonable surface modification [74]. In addition, AuNCs can be targeted for in vivo tumor transport via passive accumulation (e.g., enhanced permeability and retention effect) or active targeting (e.g., modified target molecules), thereby enhancing the bioavailability of drugs [75,76]. Moreover, the ultra-small nanostructures enable precise targeting of subcellular organelle structures, such as the nucleus and mitochondria, supplement selection, and therapeutic strategies. In combination with the unique optoelectronic and chemical properties of AuNCs, they can achieve controlled and precise strikes against the internal environmental response of tumors and external signal stimuli, consequently reducing the toxic side effects that accompany chemotherapy.

The covalent modification of the AuNP surface generally adopts sodium borohydride reduction and ligand replacement methods, and the non-covalent binding mainly includes electrostatic interaction and hydrophobic interaction to adsorb the surrounding molecules, thus reducing the surface free energy. Jiang et al. prepared AuNCs loaded with adriamycin by a “green chemistry” approach using green tea extract, in which adriamycin was co-polymerized with the nanoclusters by π-π superposition and electrostatic interactions. The drug delivery system has good stability and can significantly inhibit tumors through the synergetic effect of photothermal therapy and chemotherapy [77]. The Au-Cys-MTX/DOX NCs system constructed by Wu et al. is more stable than the non-covalent drug delivery system and can choose different drug release mechanisms according to the microenvironment of different tumor tissues. And due to the solidity of the amide bond, only a small amount of drug can be released from Au-Cys-MTX/DOX NCs in the absence of protease [78].

### 3.1. Tumor Microenvironment Response

Stimulus-responsive drug carriers are currently the hottest area of pharmacological research, especially multifunctional smart nanostructures designed for the tumor microenvironment (TME) [79]. During tumor development, TME characterized by tissue hypoxia, decreased pH, and nutrient deficiency, is formed [80]. TME is complex and multifaceted, which is involved in tumor growth, metastasis, and angiogenesis. It includes extracellular stroma and stromal cells, as well as TME metabolites and particular environments [81]. On the basis of these distinctions from normal cellular tissues, researchers have begun employing these unique microenvironments to achieve differential treatment of tumor sites.

For instance, rapid tumor growth in solid tumors increases oxygen and glucose needs. Blood supply to tumor tissue is insufficient, leading to hypoxic TME. Due to insufficient oxygen supply, tumor cells only use anaerobic enzymes for energy metabolism. Additionally, tumor tissues have an incomplete vascular system, so decomposition products cannot be eliminated in time, causing lactic acid accumulation. Various ion exchangers on the tumor cell membrane transport H^+^ produced in the cells to the outside to prevent self-acidosis. These cellular reactions lower the tumor microenvironment’s pH and acidify the TME [82,83,84]. Srinivasulu et al. developed a novel functional nanocomposite capable of responsive drug release behavior to internal environmental pH, consisting of atomically precise AuNCs (Au_22_NCs) and biopolymers (i.e., chitosan) coupled with chemotherapeutic platinum (Pt (IV)) pre-drugs and photodynamic aminolevulinic acid (ALA) pre-drugs. The acidic pH environment was able to trigger the nanocomposite for effective drug release. In contrast, drug release was inhibited at pH = 7.4 and its hydrodynamic diameter could remain constant for 7 days. It also showed a significant killing effect on A549 under synergistic light conditions, but no cytotoxicity to the normal lung cell line (MRC 5) in dark environment [85].

Latorre et al. constructed an albumin-stabilized gold nanocluster (AuNCs-DS) drug delivery system modified by doxorubicin (DOX) and SN38 for multidrug combination chemotherapy. When exposed to the low pH of the tumor microenvironment, it induces the release of DOX from the drug delivery system, while the release of SN38 is caused by a reducing environment in the cell, such as excess glutathione [86]. The concentration of glutathione (GSH) in tumor cells is 7–10 times higher than that in normal cells, and the concentration of GSH in normal cells is 200 times higher than that in extracellular cells, thus making the tumor cells a highly reducing environment.

Sun et al. took full advantage of the activity of legumain, a protease endogenously overexpressed in tumor cells, and attached its substrate fragment alanine–alanine–asparagine (AAN) peptide to AuNCs to form a larger agglomerated structure, designated “nanocluster-bomb” (Figure 3). When the AuNC-based aggregates with modified target molecules precisely reach the tumor cells, the peptide hydrolysis initiated by the highly active legumain will further trigger the disintegration of the “nanocluster-bomb”, turning on the fluorescent signal switch and releasing the chemotherapeutic and gene therapy drug molecules, achieving contrast-enhanced cancer imaging and efficient gene/chemo combination therapy [87].

### 3.2. Photoactivation

In addition to being activated by the tumor internal microenvironment, gold nanocluster-based drug delivery systems can precisely respond to external stimuli. On the surface of AuNCs, a large number of free electrons are excited to produce a collective oscillation known as surface plasma. When light is absorbed by the surface plasma, it can be converted into thermal energy, giving AuNCs exceptional photothermal conversion capability [88,89,90,91].

As mentioned above, AuNCs can be applied not only for photothermal imaging and photothermal therapy, but in fact, by responding to external light signals, they can also achieve controlled release of photosensitized drugs and exert synergistic therapeutic effects. Tabero et al. reported the preparation of a novel photopharmacological nanosystem with AuNCs of Protoporphyrin IX (PpIX) and DOX. In vitro cellular experiments, after red light irradiation of PpIX, AuNC-PpIX-DOX released DOX and synergistically-induced tumor cell death by photodynamic therapy, while this photoactive nanosystem was nontoxic in a light-free environment [92]. Yang et al. developed a multifunctional therapeutic platform (HG-AuNCs/GO-5FU, HG refers to HA-GSH) whose single-linear state oxygen production was inhibited by graphene oxide, but could be restored by lysosomal hyaluronidase in tumor cells. Meanwhile, the enzymatic degradation of hyaluronic acid (HA) on the platform and the photothermal effect of graphene oxide (GO) stimulate the release of 5-fluorouracil (5FU) from HG-AuNCs upon laser irradiation. Thus, an imaging-guided combination therapy (chemotherapy, photothermal therapy, photodynamic therapy) of HG-AuNCs/GO-5FU is realized as an anti-tumor strategy. Moreover, in in vitro cellular experiments, the complex HG-AuNCs/GO-5FU was found to have inconsistent internalization effects in two different human cancer cells (A549 and MCF-7) and normal mouse cells (CHO), attributed to their varying levels of expression of CD44, suggesting that the therapeutic platform HG-AuNCs/GO-5FU has a high selective targeting to tumor cells [93].

Photodynamic therapy (PDT), a kind of phototherapy comprising light irradiation and a photosensitizer, is extensively used in clinics for the treatment of noninvasive tumors owing to its relative non-invasiveness, greater effectiveness, and minimal side effects. Visible light activates injected photosensitizers to produce reactive oxygen species (ROS), such as singlet oxygen (^1^O_2_), which may induce phototoxicity and kill target cells [94,95]. From this viewpoint, AuNCs as drug carriers for delivering photosensitizers is now a promising option for applications combining PDT and PTT. Using AuNCs as a carrier for photosensitizers allows simultaneous activation of PTT and PDT by the same light radiation, as opposed to other methods which require the use of various excitation wavelengths of light. Along with the steady and effective photothermal conversion efficiency, high photostability and simple structure, as well as the outstanding biocompatibility of AuNCs, more and more studies have been conducted on the combined use of PTT and PDT in AuNCs, as photoactivated carriers. Liu et al. described the use of biocompatible captopril-stabilized Au nanoclusters Au_25_(Capt)_18_ for the concurrent PTT and PDT therapy of cutaneous squamous cell carcinoma (cSCC) using an 808 nm near-infrared (NIR) laser (Figure 4). Utilizing their high light-thermal conversion efficiency, potent generation of singlet oxygen, and strong photothermal stability, Au_25_(Capt)_18_ nanoclusters enhanced the suppression of cSCC XL50 cell proliferation in vitro and the inhibition of cSCC tumors on SKH-1 mice in vivo with minimal adverse effects [96].

### 3.3. Nuclear Targeting

Due to the ultra-small size smaller than the nuclear pore, AuNCs are widely used for cell nuclear-targeted transport [97]. Gao et al. found that AuNCs, synthesized by aprotinin templates, exhibited dynamically subcellular localization from the cytoplasm to the nucleus in Hela cells [98]. Tan et al. reported an AuNC cluster synthesized using an anticancer active peptide as a template, which can gradually disintegrate into separate ultrasmall AuNC particles in tumor cells and eventually localize in the nucleus and emit intense fluorescence [98,99].

Thus, using AuNCs as a carrier is a wise choice for targeted transport and precise strike on tumor cell nuclei. In 2016, Chen et al. developed a nano-platform with dual targeting capabilities by conjugating AuNC with cyclic Arg-Gly-Asp (cRGD) and aptamer (Apt) AS1411, which has a high affinity for nucleolin overexpressed in tumor cells. The AuNC-based carrier was also functionalized with NIR fluorescent dye (MPA) and DOX, a commonly used clinical chemotherapeutic medication that kills cancer cells by intercalating DNA in the cellular nucleus. In multiple tumor cell lines, tumor spheroids, and tumor-bearing mice models, the increased tumor affinity, deep tumor penetration, and enhanced anti-tumor effectiveness of this prodrug were shown. This work not only reveals the possibility of non-toxic AuNC modified with two targeting ligands for tumor-targeted imaging, but also demonstrates the bright future of dual targeting AuNC as a core for the creation of prodrugs for cancer treatment [100].

## 4. AuNCs as Therapeutic Agents in Combined Therapy

### 4.1. Radiosensitization

Radiation therapy is an effective oncology treatment that uses ionizing radiation to target cancer cells. A part of the high-energy radiation will nonetheless be transferred to the normal tissues around the tumor, inflicting irreparable damage [101,102]. Gold has a high atomic number and much greater electron density than soft tissues, which may boost photoelectric absorption and secondary electron yield, improve local energy deposition in tumor tissues, and expedite the death of tumor cells [103,104]. In addition to the previously reported physical methods, gold clusters may potentially exert their radiosensitizing effects via biological pathways, for example, controlling the cell cycle, boosting free radical generation in response to radiation, altering cell autophagy, and causing apoptosis [105,106]. Using nanogold for the first time for in vivo tumor radiation sensitization, Herold et al. demonstrated that gold particles might have a dose-enhancing impact on cells and C.B17/Icr scid mice cultured with EMT-6 mouse tumor cells when exposed to 200 kVp X-rays both in vitro and in vivo tests [107].

The enhanced permeability and retention (EPR) effect may enhance the aggregation of AuNCs in the tumor, since smaller nanoclusters (<5 nm) can more easily penetrate tumor tissues and cross blood vessels than larger nanoparticles (>10 nm). Additionally, the tumor tissue’s decreased lymphatic outflow makes it difficult for nanoparticles to be effectively cleared away, which increases their retention in the tumor tissue [108,109], hence improving the radiation treatment impact and reducing the harm to the surrounding normal cells. Attaching trastuzumab and folic acid targeting human epidermal growth factor receptor 2 (HER2) to 4.2 nm AuNCs, Roghayeh et al. demonstrated that the targeted AuNCs may infiltrate breast cancer SK-BR3 cells through HER-2-mediated mechanisms [110]. Luo et al. created therapeutic AuNCs that can function as prostate cancer (PCa)-targeted radiosensitizers and chemotherapy carriers. Using PSMA-MMAE as a template and the reduction of Au^3+^ by the reactive group, PSMA-AuNC-MMAE couples were synthesized. The gold nanocluster-attached prostate-specific membrane antigen (PSMA) could improve the targeting of AuNCs; the bound monomethyl auristatin E (MMAE) was a chemotherapeutic prodrug that enhanced the chemotherapeutic effect after binding with AuNC, and MMAE could also boost the radiosensitizing impact by inhibiting the cells in the G2-M area. Due to PSMA receptor amplification, PC3pip tumor cells maintained considerably more gold nanocluster complexes in PC3pip tumor-bearing animals than in PC3flu tumor-bearing mice, as shown by in vivo tests [111]. Wu et al. created transformable gold nanocluster (AuNC) aggregates (called AuNC-ASON) using antisense oligonucleotides (ASON) that target survivin mRNA. The acidic tumor microenvironment modifies the electrostatic interactions between the polyelectrolyte poly(allylamine) (PAH) and glutathione surface ligands that stabilize AuNC, causing gold nanocluster aggregates to separate into 2 nm AuNCs and triggering the release of loaded antisense oligonucleotides for gene silencing. The findings of in vivo tests analyzing the co-localization of AuNC-ASON with nucleosomes/lysosomes revealed that AuNC-ASON has an excellent nucleosome/lysosome escape capacity. Moreover, real-time polymerase chain reaction (real-time PCR) research revealed that the expression of survivin mRNA in 4T1 cells followed the same pattern as cell viability, validating the mechanism of tumor cell eradication based on survivin gene silencing. With the aid of survivin gene interference, this treatment approach may increase and enhance the radiosensitivity of cancer cells and enable the simultaneous use of tumor radiation and gene therapy [112].

The effectiveness of radiotherapy is contingent upon radiosensitivity, and a hypoxic tumor microenvironment renders tumor cells more resistant to ionizing radiation. As radiosensitizers, AuNCs may be used with oxygen carriers to reduce tumor hypoxia by generating reactive oxygen species (ROS) generation and enhancing the effectiveness of radiation. In the cRGD multifunctional treatment system, Au_4_-IO NP-cRGD triggered the death of 4T1 cells by producing substantial quantities of reactive oxygen species in response to X-ray exposure. Experiments in vivo have shown that this multifunctional treatment platform is capable of directing Fenton response-assisted improved radiotherapy using dual-mode imaging based on magnetic resonance imaging of iron oxide (IO) nanoclusters and fluorescence imaging of Au_4_ clusters [113].

Due to the combination of physical, chemical, and biological factors, radiosensitization is a complicated phenomenon [114]. The processes by which AuNCs exhibit radiosensitizing effects, particularly the biological pathways involved, are not well understood. In addition, the efficacy of the functionalized modification of AuNCs targeted to tumor tissues when administered in vivo, as well as the harm to healthy tissues and non-specific accumulation of long-term damage to persons, need more research. In the meantime, the increase in the size of AuNCs after different surface modifications reduces the clearance rate in the organism and increases the accumulation in the liver. However, it has not been conclusively determined whether there is an influence on the gene expression of individuals.

In fact, tumor tissues grow at inconsistent rates in all directions with irregular edges. Whether AuNCs can be conformally distributed according to the different shapes of tumors needs to be further investigated. In conclusion, more animal experiments and preclinical trials are needed for the practical translation of AuNCs to ensure that the multifunctional system of AuNCs can be efficiently and safely applied in the clinic.

### 4.2. Photothermal Conversion

As we mentioned earlier, the excellent photothermal conversion efficiency of AuNCs allows them to be used as ideal photothermal agents for multimodal imaging and therapeutic implementation. Therefore, examples of combined treatment based on the photothermal action of AuNCs will not be repeated in this section.

Focusing on the optimization aspect of the photothermal effect of AuNCs, recently, Yin’s group developed AuNCs as highly efficient photothermal treatment agents and provided a semiquantitative technique for determining their resonant frequency and absorption efficiency by integrating practical medium approximation theory with full-wave electrodynamic simulations. Guided by this theory, they created a space-confined seeded growth approach to prepare AuNCs. Under optimum growth circumstances, they obtained a record photothermal conversion efficiency of 84% for gold-based nanoclusters, due to collective plasmon-coupling-induced near-unity absorption efficiency. They showed the improved exceptional photothermal treatment performance of AuNCs in vivo. Their study shows the potential and effectiveness of AuNCs as nanoscale photothermal treatment agents [115].

Lately, a nanoarchitecture comprising a conjugated gold nanorod (AuNR) and gold cluster hybrid system was developed to optimize the photothermal conversion efficiency. Due to the target specificity of folate receptors for cancer cells, the hybrid material exhibited high in vitro therapeutic efficacy after folic acid conjugation. More importantly, the nanoarchitecture of the hybrid material had no significant influence on the optical and thermal properties of either AuNCs or AuNRs, but exerted enhanced photothermal effects [116].

## 5. Conclusions

Due to worldwide environmental deterioration, the rising prevalence and lethality of malignant tumors pose a growing risk to human life and health. Moreover, the present comparatively tardy diagnosis and single therapy in clinical practice are insufficient to satisfy people’s urgent need for tumor management. Consequently, AuNCs with great potential for multimodal diagnostic and multifunctional therapeutic applications are gaining growing interest. In this review, we systematically analyzed the unique opto-electronic and physicochemical properties of AuNCs and outline the recent novel applications based on AuNCs in tumor theranostic and combination therapy (Table 1). Here, the potential and remaining shortcomings of AuNCs for multifunctional applications are categorized and discussed in terms of their three major roles in practical applications, including imaging agents, drug carriers, and therapeutic nanosystems.

**Table 1 pharmaceutics-14-02451-t001:** Application of AuNCs.

Multifunctional Nanoplatform	Role of AuNCs	Therapeutic Agent	Size (nm)	Imaging Mode	Cancer Types	Therapy Method	Activity	Ref
AuNCs-Ag@Keratin-Gd	Imaging	NM	5	FL, MRI	Breast cancer	Chemotherapy	In vivo and in vitro	[37]
CDGM NPs	Imaging, drug delivery	CAD, Ce6	2	FL	Lung cancer	PDT	In vivo and in vitro	[54]
AuS-U11	PTT-carrier	U11 peptide, cyanine dye Cy5.5, 5-ALA	10	FL	Pancreatic carcinoma	PTT, PDT	In vivo and in vitro	[55]
Au NBPs@PDA/AuNCs	Imaging	Au NBPs@PDA	2.1, 3.3	FL	Breast cancer, hepatocarcinoma	PTT	In vitro	[56]
Dox@HG-CAHs	Imaging	HA-ALD, Dox	2.8	FL, CT	Osteosarcoma	PTT, chemotherapy	In vivo and in vitro	[61]
AuNCs–LHRHa	Imaging, PTT	LHRH analogues	2.4	FL, CT	Prostatic cancer	PTT	In vitro	[67]
GTSL-CYC-HER2	Changed the zeta potential of liposomes, superior photothermal effect	HER2-modified thermosensitive liposome, cyclopamine	NA	CT, PTI	Breast cancer	Chemotherapy, PTT	In vivo and in vitro	[68]
Ce6&AuNCs/Gd-LDH	Imaging	Ce6	~2	MRI, FL	Hepatocarcinoma	PDT	In vivo and in vitro	[70]
AuNCs-ICG	Imaging, radiosensitizing effects	ICG	~1	FL, PAI, CT	Breast cancer	PDT, RT	In vivo and in vitro	[72]
Qu-GNCs	Imaging	Qu	1–3	FL	Lung cancer	Chemotherapy	In vitro	[74]
Fe3O4@PAA/AuNCs/ZIF-8 NPs	Imaging	DOX	NA	MRI, CT, FL	Hepatocarcinoma	Chemotherapy	In vivo and in vitro	[75]
AuNCs@GTMS-FA	Imaging, phototherapeutic agents	FA	2.8	FL	Breast cancer	PTT, PDT	In vitro	[80]
AuNCs/Dzs-Dox	NSET effect, shelter therapeutic cargos	Dzs-Dox	~1.76	FL	Breast cancer	Gene therapy, chemotherapy	In vivo and in vitro	[87]
HG-GNCs/GO-5FU	Bioimaging, phototherapeutic	HA, 5FU	2	FL	Lung cancer, breast cancer	Chemotherapy, PDT, PTT	In vitro	[93]
AuNCs@mSiO_2_@MnO_2_	Photosensitizer	MnO_2_ nanozyme	NA	MRI	Breast cancer	PDT	In vivo and in vitro	[94]
Au8NC	Radiosensitizing effects	Levonorgestrel	~2	FL	Esophagus cancer	RT	In vivo and in vitro	[105]
Au_4_-IO NP-cRGD	Imaging, radiosensitizing effects	IO nanocluster	2	FL, MRI	Breast cancer	RT, chemotherapy	In vivo and in vitro	[113]
PML-MF nanocarrier	Imaging	IO@AuNPs	NA	FL	Cervical cancer	PPTT, chemotherapy	In vitro	[117]
WLPD-Au_25_	Photosensitizer, drug delivery	WS2 nanoparticles, Dex, Captopril	2.5	CT	Breast cancer	PTT, PDT	In vivo	[118]
AuNCs/Cas9–gRNA	Imaging, drug delivery	Cas9–sgRNA plasmid	~1.56	FL	Osteosarcoma	Gene therapy	In vitro	[119]
K-AuNCs	Imaging, drug delivery	K	1–3	FL	Lung cancer	Chemotherapy	In vitro	[120]
EA-AB	Imaging	EB	NM	FL, MSOT Imaging	Breast cancer	Chemotherapy, PTT	In vivo and in vitro	[121]
Ce6-GNCs-Ab-CIK	Drug delivery	Ce6, CD3 antibody	NA	FL	Gastric cancer	Chemotherapy, PDT	In vivo and in vitro	[122]
Au_4_Cu_4_/Au_25_@Lip	Photothermogenesis effect, photoluminescence performance	Au_4_Cu_4_ nanoclusters	~2	FL, PTI	Cervical cancer	PTT, PDT	In vivo and in vitro	[123]
MB-loaded Au NC-mucin NPs	Imaging	MB	1.9 ± 0.34	FL	Cervical cancer	PDT	In vitro	[124]
ISQ@BSA-AuNC@AuNR@DAC@DR5	SERS substrate	DAC, ISQ	NA	NM	Amelanotic Melanoma	PTT, PDT	In vivo and in vitro	[125]

Abs: PTT: photothermal therapy; NSET: nanosurface energy transfer; CAD: MMP2 polypeptidecis-aconitic anhydride-modified doxorubicin; Ce6: photosensitizer chlorin e6; Au NBPs@PDA: polydopamine-capped gold nanobipyramids; HA-ALD: oxidized hyaluronic acid; Dox: doxorubicin; Ce6: chlorin e6; ICG: indocyanine green; K: Kaempferol; Qu: Quercetin; EB: Erlotinib; HA: hyaluronic acid; 5FU: 5-fluorouracil; FA: Folic acid; MB: Methylene blue; DAC: Dacarbazine; Dzs-Dox: DNAzyme, Dox; IO: Iron oxide; Dex: Dexamethasone; FL: fluorescence imaging; MRI: magnetic resonance imaging; CT: computed tomography; PT: photothermal imaging; PAI: photoacoustic imaging, MSOT: multispectral optoacoustic tomography; PDT: photodynamic therapy; RT: radiation therapy; PPTT: plasmonic photothermal therapy; NM: not mentioned; NA: not applicable.

Although AuNCs have come a long way since they were first made, they are still evolving quickly in their designs and applications in biomedicine. For AuNCs, there are still a several obstacles to overcome in their future development. The first is that the synthesis products of AuNCs cannot be properly regulated artificially, especially when functionalized on their surfaces with template peptides and proteins. In order to properly regulate the finished synthesized products, therefore, it is necessary to understand the characteristics of AuNCs and the mechanism of their production [126,127,128]. Secondly, there are still many questions regarding the mechanisms of action that need to be further addressed before AuNCs can be employed in clinical practice. Current research on AuNCs is conducted more at the cell and experimental animal level, lacking comprehensive biological effects and mechanisms of AuNCs in vivo. Thirdly, there is no recognized standard for the accurate biodistribution of AuNCs in the body, and there is no reported evidence on whether they are completely eliminated from the body. These factors have resulted in the consistent inability to develop a uniform standard to adequately assess the toxicity of AuNCs. Finally, AuNCs are costly to synthesize and difficult to commercially produce on a large scale. Therefore, how to optimize the synthesis process and reduce cost make fundamental prerequisites to advance their clinical translation.

In conclusion, it is hoped that this review can provide experience and inspiration for researchers in this field and promote the further development and clinical translation of AuNCs.

## Data Availability

Not applicable.

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
