# Peer review of "The Recent Development of Multifunctional Gold Nanoclusters in Tumor Theranostic and Combination Therapy"

_pharmaceutics, 2022, doi:10.3390/pharmaceutics14112451_

Round 1

Reviewer 1 Report

The review entitled "The recent development of multifunctional gold nanocluster in therapeutic therapy and combined tumor" offers an overview of the use of gold nanoclusters (AuNCs) in combined therapy and theranostics of cancer. The manuscript describes the potential of AuNCs as contrast agents, therapeutic agents, and adjuvants in the treatment of cancer, giving some interesting examples from recent literature.

The topic is interesting, and the manuscript is mostly pleasant to read. However, the reviewer suggests some changes to improve the value of the manuscript:

1)      In the Abstract and Introduction sections, there are some typographical errors. Some examples: in the abstract, smaller and fewer instead of small and few; in the keywords, diagnose instead of diagnosis; in the first line of the introduction the verb pose should be in the plural form; more important, in the fifth line of page 2, atomic number instead of number of atoms (this could be misleading). Other typos occur sporadically here and there. Please, check the text carefully.

2)      The work described in section 2.1 on the use of gold nanobipyramids is undoubtedly interesting and inspiring, but it does not seem to be about the use of AuNCs. Please specify whether AuNCs have been used or otherwise the reference is not considered relevant and thus should be removed.

3)      The inclusion of a table that summarizes the most interesting uses of AuNCs in cancer theranostics would greatly increase the descriptive value of the review. For this reason, it is strongly suggested to insert a table related to Chapter 2 in which the examples described (and possibly further examples) are outlined in order to highlight the main features. A possible example of a table could show the type of theranostic system, the role of AuNCs, the therapeutic agent (other than AuNCs, if any), the type of imaging technique, the cancer type, and in vitro or in vivo therapeutic activity.

4)      In section 4.1 the authors state that “The enhanced permeability and retention (EPR) effect may enhance the aggregation of gold nanoclusters in the tumor, hence improving the radiation treatment impact and reducing the harm to the surrounding normal cells”. It is not clear the meaning of “enhanced aggregation” due to EPR effect. Please, clarify this point and add a reference that refers to the size of the nanoclusters under consideration.

5)      The section on the limitations of AuNCs therapy should be expanded with examples of recent literature that could provide a more solid framework which would greatly help researchers wishing to undertake this line of research.

6)      In the revised version it is strongly recommended to use line numbers as reported in the template. This would help the reviewer to provide more detailed suggestions.

Author Response

Response to Reviewer 1 Comments

Point 1: In the Abstract and Introduction sections, there are some typographical errors. Some examples: in the abstract, smaller and fewer instead of small and few; in the keywords, diagnose instead of diagnosis; in the first line of the introduction the verb pose should be in the plural form; more important, in the fifth line of page 2, atomic number instead of number of atoms (this could be misleading). Other typos occur sporadically here and there. Please, check the text carefully.

Response 1: Thank you very much for your valuable suggestion. The full text has been carefully reviewed and in revision mode, including, but not limited to, changes made in response to the queries you have raised.

Point 2: The work described in section 2.1 on the use of gold nanobipyramids is undoubtedly interesting and inspiring, but it does not seem to be about the use of AuNCs. Please specify whether AuNCs have been used or otherwise the reference is not considered relevant and thus should be removed.

Response 2: Thank you for pointing that out. The reference utilized dual-mode GNC probes for labeled imaging of two biomolecules. To specify the important roles of AuNCs in this work, we have adjusted our expressions as “ For simultaneous in situ multilayer imaging with differing intracellular spatial distribu-tions and fluorescence-guided photothermal treatment, Wang et al. have created a multi-functional nanomaterial-based all-in-one nanoplatform combining polydopamine (PDA) loaded gold nanobipyramids and two different kinds of AuNC probes (Figure 1). In this elaborate design, AuNCs with different emission wavelengths were synthesized and tagged with 2 molecular probes, transmembrane glycoprotein mucin1 (MUC1) aptamer and microRNA-21, respectively. The MUC1 aptamer labeled AuNCs detached from the nanocomposites upon the competitive combinations between MUC1 and its aptamer, turning on red fluorescence. Inside the cells. miRNA-21 hybridization with sin-gle-stranded DNA caused green fluorescence in AuNCs.” in section 2.1.

Point 3: The inclusion of a table that summarizes the most interesting uses of AuNCs in cancer theranostics would greatly increase the descriptive value of the review. For this reason, it is strongly suggested to insert a table related to Chapter 2 in which the examples described (and possibly further examples) are outlined in order to highlight the main features. A possible example of a table could show the type of theranostic system, the role of AuNCs, the therapeutic agent (other than AuNCs, if any), the type of imaging technique, the cancer type, and in vitro or in vivo therapeutic activity.

Response 3: Thanks for your valuable suggestion.“Table 1. Recent multifunctional application of AuNCs in tumor theraostic and combination therapy” has been added to Conclusion in Section 5 , given that the contents of the table include all the recent relevant reports covered in this manuscript. 

Point 4: In section 4.1 the authors state that “The enhanced permeability and retention (EPR) effect may enhance the aggregation of gold nanoclusters in the tumor, hence improving the radiation treatment impact and reducing the harm to the surrounding normal cells”. It is not clear the meaning of “enhanced aggregation” due to EPR effect. Please, clarify this point and add a reference that refers to the size of the nanoclusters under consideration.

Response 4: Thank you for pointing that out. As you suggested, the corresponding explanation of EPR has been revised and relevant references have been added.

Point 5: The section on the limitations of AuNCs therapy should be expanded with examples of recent literature that could provide a more solid framework which would greatly help researchers wishing to undertake this line of research.

Response 5: Thanks for your valuable suggestion. The suggested analysis and conclusion of the limitations of AuNCs has added in the Section 5 of Conclusion based on several references.

Point 6: In the revised version it is strongly recommended to use line numbers as reported in the template. This would help the reviewer to provide more detailed suggestions.

Response 6: Thank you for pointing that out. Line numbers have been set in the revised manuscript.

Reviewer 2 Report

The authors present a detailed study on the unique opto-electronic
and physico-chemical properties of gold nanoclusters and outline the recent novel appli-
cations based on gold nanoclusters in tumor theranostic and combination therapy.

The review is well organized. However, before to be published some manadtory changes must be done.

The authors should use the acronyms AuNPs or AuNCs any time after introduction.

The references are not complete. It is obvious that an fastly increasing field as theranostics could present a problem for inclusing the relevant papers. Howver I suggest to include the following paper:

Detection of intracellular nanoparticles. An overview

Mario D'Acunto, Materials, 18 , 2018 11(6), 882; https://doi.org/10.3390/ma11060882

After such changes the paper could be reconsidered for a publication.

Author Response

Response to Reviewer 2 Comments

Point 1: The authors should use the acronyms AuNPs or AuNCs any time after introduction.

Response 1: Thank you for pointing that out. After careful examinations, we made careful corrections to all abbreviations in this revised manuscript.

Point 2: The references are not complete. It is obvious that an fastly increasing field as theranostics could present a problem for inclusing the relevant papers. Howver I suggest to include the following paper:

Detection of intracellular nanoparticles. An overview

Mario D'Acunto, Materials, 18 , 2018 11(6), 882; https://doi.org/10.3390/ma11060882

Response 2: Thanks for your suggestion. The suggested references have been properly cited in Content 3.2 of the revised manuscript.

Reviewer 3 Report

In this review paper, the authors presented "The recent development of multifunctional gold-nanoclusters in tumor theranostic and combination therapy". From my point of view, the topic is fascinating. The paper is concise. However, some issues need to be addressed before its publication in Pharmaceutics. Following are my suggestions:

1)      The headings and subheadings look confusing. It makes the article difficult to understand. The authors should change the heading and subheadings of the paper to be small and crisp so anyone can easily understand it. It is not necessary to include all the details in the headings.

2)      Page 4, subheading 2.2 and paragraph second: Authors need to check the following sentence carefully: “Although the advantages of smaller gold nanoclusters in radiography are not as advantageous as those of larger gold nanoparticles, the local plasmon resonance and larger cavity structure caused by the ultra-small size give AuNCs a greater photothermal con-version effect, allowing them to be used for both photothermal imaging and photothermal therapy58, 59.”  Authors, please cite the relevant and specific articles where local plasmon resonance and large cavity structure have been discussed. 58 and 59 are the review articles. 

Author Response

Response to Reviewer 3 Comments

Point 1: The headings and subheadings look confusing. It makes the article difficult to understand. The authors should change the heading and subheadings of the paper to be small and crisp so anyone can easily understand it. It is not necessary to include all the details in the headings.

Response 1: Thanks for your valuable suggestion. As you suggested, we have carefully considered and revised the heading and subheadings of the chapters in revised manuscript as follows:

  1. AuNCs as imaging agents in tumor theranostic

          2.1. Fluorescence imaging

         2.2. Radiography

         2.3. Multi-modal imaging

  1. AuNCs as transport agents in combined therapy

         3.1. Tumor microenvironment response

         3.2. Photoactivation

         3.3. Nuclear targeting

  1. AuNCs as therapeutic agents in combined therapy

         4.1. Radiosensitization

         4.2. Photothermal conversion

Point 2: Page 4, subheading 2.2 and paragraph second: Authors need to check the following sentence carefully: “Although the advantages of smaller gold nanoclusters in radiography are not as advantageous as those of larger gold nanoparticles, the local plasmon resonance and larger cavity structure caused by the ultra-small size give AuNCs a greater photothermal con-version effect, allowing them to be used for both photothermal imaging and photothermal therapy58, 59.”  Authors, please cite the relevant and specific articles where local plasmon resonance and large cavity structure have been discussed. 58 and 59 are the review articles.

Response 2: Thanks for pointing that out. The related original articles have been properly cited in revised manuscript.

Reviewer 4 Report

The article by Liu et al.  reviews the development and use of metal nanoclusters as theranostic tools and in combination therapy. The article is well-written and a comprehensive view of gold nanoclusters’ relevance in the biomedical field is provided. I am able to recommend the publication of the article in Pharmaceuticsupon completion of a few minor revisions:

1)    In the introduction some relevant recent literature about gold nanoclusters in biomedical applications should be cited:

-       Luminescent gold nanoclusters for bioimaging applicationsBeilstein Journal of Nanotechnology, 42, 533 - 546

-       Development of gold nanoclusters: from preparation to applications in the field of biomedicine, J. Mater. Chem. C, 2020,8, 14312-14333

2)    In the introduction section, a Figure/Scheme summarizing the main features of gold nanoclusters and their method of preparation would help the reader.

3)    The article is quite long, with many paragraphs. A short description of the paragraphs organization and of the criteria chosen to organize the review should be provided to guide the reader.

4)    Beyond fluorescent properties, Gold Nanoclusters are also endowed with exceptional self-assembly and crystallogenic features, owing to their precise size and composition. The authors should briefly mention this and cite relevant recent literature including:

-       High-Resolution Crystal Structure of a 20 kDa Superfluorinated Gold Nanocluster, Nature Communications 2022 13 (1), 1-8

-       From Precision Colloidal Hybrid Materials to Advanced Functional Assemblies,  Acc. Chem. Res. 2022, 55, 13, 1785–1795

5)    The article tells many successful stories in which gold nanoclusters have shown to hold promise. A discussion highlighting the challenges and limitations of these systems and how to address them would certainly enrich the review.

Author Response

Response to Reviewer 4 Comments 

Point 1: In the introduction some relevant recent literature about gold nanoclusters in biomedical applications should be cited:

Luminescent gold nanoclusters for bioimaging applications, Beilstein Journal of Nanotechnology, 42, 533 - 546

Development of gold nanoclusters: from preparation to applications in the field of biomedicine, J. Mater. Chem. C, 2020,8, 14312-14333

Response 1: Thanks for your valuable suggestion. The suggested references have been added that point to the relevant textual material.

Point 2: In the introduction section, a Figure/Scheme summarizing the main features of gold nanoclusters and their method of preparation would help the reader.

Response 2: Thanks for your valuable suggestion. The Scheme containing the main features, preparation and applications of AuNCs have been added in the Introduction as you suggested.

Point 3: The article is quite long, with many paragraphs. A short description of the paragraphs organization and of the criteria chosen to organize the review should be provided to guide the reader.

Response 3: Thanks for your valuable suggestion. The description of the paragraphs organization and of the criteria chosen to organize the review have been added in the end of Introduction in the revised manuscript as “Hence, this review focuses on the use of AuNCs as imaging, transport, and therapeutic agents for combined application. Specifically, in Section 2, we summarized the applica-tions of AuNCs in fluorescence imaging, radiography and other multimodal imaging for tumor theranostic. In Section3, strategies for controlled release via tumor microenviron-ment response, photoactivation and nuclear targeting based on using AuNCs in con-structing smart multifunctional vectors were discussed. Finally in Section 4, the combined therapies for cancer utilizing the radiosensitization and photothermal conversion effects of AuNCs were highlighted.”.

Point 4: Beyond fluorescent properties, Gold Nanoclusters are also endowed with exceptional self-assembly and crystallogenic features, owing to their precise size and composition. The authors should briefly mention this and cite relevant recent literature including:

High-Resolution Crystal Structure of a 20 kDa Superfluorinated Gold Nanocluster, Nature Communications 2022 13 (1), 1-8

From Precision Colloidal Hybrid Materials to Advanced Functional Assemblies,  Acc. Chem. Res. 2022, 55, 13, 1785–1795

Response 4: Thanks for your valuable suggestion. The discussion about the self-assembly and crystallogenic features of AuNCs have been added in the Section 2, and the relevant literature has been cited accordingly as you suggested.

Point 5: The article tells many successful stories in which gold nanoclusters have shown to hold promise. A discussion highlighting the challenges and limitations of these systems and how to address them would certainly enrich the review.

Response 5: Thanks for your valuable suggestion. The suggested analysis and conclusion of the limitations of AuNCs has added in the Section 5 of Conclusion based on several references.